# Role of Prosaposin and Extracellular Sulfatase Sulf-1 Detection in Pleural Effusions as Diagnostic Biomarkers of Malignant Mesothelioma

**DOI:** 10.3390/biomedicines10112803

**Published:** 2022-11-03

**Authors:** Lorenzo Zallocco, Roberto Silvestri, Federica Ciregia, Alessandra Bonotti, Riccardo Marino, Rudy Foddis, Antonio Lucacchini, Laura Giusti, Maria Rosa Mazzoni

**Affiliations:** 1Department of Pharmacy, University of Pisa, 56100 Pisa, Italy; 2Department of Biology, University of Pisa, 56100 Pisa, Italy; 3Occupational Medicine Unit, Department of Translational Research and New Medical and Surgical Technologies, University-Hospital of Pisa, 56100 Pisa, Italy; 4Department of Clinical and Experimental Medicine, University of Pisa, 56100 Pisa, Italy; 5School of Pharmacy, University of Camerino, 62032 Camerino, Italy

**Keywords:** mesothelioma, pleural effusion, lung adenocarcinoma, biomarker, prosaposin, extracellular sulfatase Sulf-1

## Abstract

Malignant pleural mesothelioma is an aggressive malignancy with poor prognosis. Unilateral pleural effusion is frequently the initial clinical sign requiring therapeutic thoracentesis, which also offers a diagnostic opportunity. Detection of soluble biomarkers can support diagnosis, but few show good diagnostic accuracy. Here, we studied the expression levels and discriminative power of two putative biomarkers, prosaposin and extracellular sulfatase SULF-1, identified by proteomic and transcriptomic analysis, respectively. Pleural effusions from a total of 44 patients (23 with mesothelioma, 8 with lung cancer, and 13 with non-malignant disease) were analyzed for prosaposin and SULF-1 by enzyme-linked immunosorbent assay. Pleural effusions from mesothelioma patients had significantly higher levels of prosaposin and SULF-1 than those from non-malignant disease patients. Receiver-operating characteristic (ROC) analysis showed that both biomarkers have good discriminating power as pointed out by an AUC value of 0.853 (*p* = 0.0005) and 0.898 (*p* < 0.0001) for prosaposin and SULF-1, respectively. Combining data ensued a model predicting improvement of the diagnostic performance (AUC = 0.916, *p* < 0.0001). In contrast, prosaposin couldn’t discriminate mesothelioma patients from lung cancer patients while ROC analysis of SULF-1 data produced an AUC value of 0.821 (*p* = 0.0077) but with low sensitivity. In conclusion, prosaposin and SULF-1 levels determined in pleural effusion may be promising biomarkers for differential diagnosis between mesothelioma and non-malignant pleural disease. Instead, more patients need to be enrolled before granting the possible usefulness of these soluble proteins in differentiating mesothelioma pleural effusions from those linked to lung cancer.

## 1. Introduction

Malignant pleural mesothelioma (MPM) is a rare and extremely aggressive malignancy arising from pleural mesothelial cells [1,2]. Since 1960 the association between asbestos-exposure and MPM is well-documented with a long latency period, up to 30–60 years [3].

Disease symptoms are usually insidious and nonspecific causing diagnostic difficulties at early stages although modern imaging techniques and advances in immunohistochemical staining have lately improved diagnosis [2]. Nevertheless, MPM rapid progression and invasiveness as well as limited therapeutic improvements still lead to a dismal prognosis with a median survival time of 9–12 months from diagnosis. Prognosis mainly depends on neoplasm size, stage, and histological type but gender and age also play a role in affecting survival. Unilateral pleural effusion (PE) is frequently one of the initial clinical signs of MPM requiring therapeutic thoracentesis to alleviate patient symptoms. In addition, the PE obtained by a minimally invasive procedure allows an early diagnostic opportunity by combining cytological analysis with detection of immunocytochemical, soluble protein and nucleic acid biomarkers [4,5].

Soluble biomarkers can be found in different body fluids such as plasma, serum, and PE, which are easy to obtain with minimal patient discomfort and feasibly provide numerous information about the tumor. Furthermore, they should ideally allow to diagnose the disease at early stage, sooner than detection by imaging techniques [6]. Numerous protein biomarkers for MPM detection have been proposed and assessed for their diagnostic accuracy in serum, plasma, and PE. Mesothelin (MSLN), a plasma membrane glycoprotein, which is highly expressed in MPM cells, is the most widely studied biomarker. MSLN soluble derivatives, megakaryocyte potentiating factor (MPF) and soluble mesothelin-related peptide (SMRP), have been investigated as screening, diagnostic, and prognostic biomarkers [7]. However, serum SMRP usefulness as early diagnostic biomarker is hampered by the low sensitivity [8]. A recent systematic review examining the clinical utility of various biomarkers has concluded that PE SMRP shows the best diagnostic effectiveness while fibulin-3, which has been presented as an accurate plasma biomarker, cannot be considered useful in PE due to contradictory results reported in literature [9]. Other soluble proteins such as CYFRA-21-1, a fragment of cytokeratin 19, and hyaluronic acid have been also examined as potential MPM biomarker in PE although none shows an outstanding diagnostic accuracy [9]. Biomarker panels have been proposed as early diagnostic tools and two of them, which include SMRP, have shown better performances than single biomarkers in PE [10,11].

The pursuit for biomarkers is still ongoing with the aim to detect novel molecules with high diagnostic and prognostic value. We have interrogated the proteome of MPM tissue specimens and cell lines in order to identify novel putative biomarkers [12,13,14]. Some differentially expressed proteins in MPM tissues could be also found in serum with a fear power to discriminate between patients and asbestos-exposed healthy subject and utility to monitor disease progression [15,16]. Secretome analysis of two MPM cell lines has allowed us to identify two soluble proteins, prosaposin (PSAP) and quiescin Q6 sulfhydryl oxidase 1 (QSOX1), which are detectable in serum and show a fear/good diagnostic accuracy in distinguishing MPM patients from exposed subjects [14]. Moreover, these studies point out that biomarker panels improve the diagnostic potency by increasing both sensitivity and specificity. Other potential MPM biomarkers have been also suggested by transcriptomic analysis of tissue specimens and cell lines [17]. In this study, the transcript for the extracellular sulfatase SULF-1 (SULF-1), which modifies heparan sulfate and can influences cancer growth and spread, has been found overexpressed in MPM [17,18].

The present study was undertaken to extend the quantitative detection of new putative MPM biomarkers to PE, which is the closest body fluid to neoplasm as well as an early sign. The levels of PSAP and SULF-1 were measured in PE of MPM and control (lung cancer and benign pleural disease) patients by enzyme-linked immunosorbent assay (ELISA). The diagnostic accuracy of each biomarker alone and combined was also evaluated.

## 2. Materials and Methods

### 2.1. Patients

PEs from 44 patients, 23 MPM patients, 8 patients with lung adenocarcinoma (ADC), and 13 patients with benign pleural disease (BPD), were prospectively collected by thoracentesis at the University-Hospital of Pisa (Italy) from 2004 to 2008. When possible, a preliminary diagnosis was made by cytological analysis. The definitive diagnosis was made based on hematoxilin and eosin staining and immunohistochemistry of biopsy or surgical specimen. MPM samples were all classified as epithelioid subtype by histologic examination. All PE samples were collected before any treatment was given. Within 30 min after pleural fluid collection, samples were centrifuged and resulting supernatants were stored at −80 °C until the time of use. The study was approved by the Local Ethical Committee (Comitato per la Sperimentazione Clinica dei Farmaci, Azienda Ospedaliera-Universitaria Pisana). Table 1 summarizes the demographic characteristics of patients as well as smoking habit and asbestos-exposure.

### 2.2. Enzyme-Linked Immunoassay of PSAP and SULF-1

PSAP and SULF-1 concentrations in PE samples were measured using commercially available sandwich-type ELISA kits, following manufacturer’s instructions. ELISA kit for PSAP was from Cloud-Clone Corp. (Katy, TX, USA) (cat n. SEC756Hu) while that for SULF-1 was from Cusabio (Houston, TX, USA) (cat. n. CSB-EL022930Hu). Briefly, PE samples were diluted 1:50 (PSAP) or 1:5 (SULF-1) using 10 mM phosphate buffer saline (PBS), pH 7.4. Blanks, diluted standards and samples (100 μL) were added to 96-well plates pre-coated with an anti-PSAP or -SULF-1 antibody and incubated for 1 h at 37 °C. Then, wells were washed, an anti-PSAP or -SULF-1 biotin-conjugated antibody added, and incubated for 1 h at 37 °C. After washing, the avidin conjugated horseradish peroxidase (HRP) was added and incubated for 30 min at 37 °C. Then, plates were washed, the chromogen solution was added and incubated for 10 min at 37 °C. After that, the enzyme-substrate reaction was quenched, and color optical density (OD) was measured spectrophotometrically. PSAP and SULF-1 concentrations (ng/mL) were determined by comparing ODs of PE samples to PSAP and SULF-1 standard curves, respectively.

### 2.3. Statistical Analysis

Measurements of biomarker concentration were performed in duplicate and repeated three times for each PE sample. Data were analyzed by Shapiro-Wilk test to assess the normal distribution. Since variables were not normally distributed, differences between groups were analyzed by Mann Whitney-Wilcoxon test. To identify the predictive power of PSAP and SULF-1 to differentiate between MPM and controls (BPD and lung ADC), receiver-operating characteristic (ROC) analysis was performed. The area under curve (AUC) was calculated with 95% confidence intervals (95% CI). To estimate whether biomarker combination could increase their performance in discriminating between MPM and BPD, data were combined using a logistic regression analysis. Calculations were performed using GraphPad Prism 8.0 (GraphPad Inc., San Diego, CA, USA) and SPSS Statistics 20.0 (SPSS Inc., Chicago, IL, USA).

## 3. Results

### 3.1. Expression Levels of PSAP and SULF-1 in MPM, BPD, and Lung ADC Pleural Effusion

In order to evaluate the diagnostic power of PE derived PSAP and SULF-1, we compared the expression level of these putative biomarkers in pleural fluids from MPM (*n* = 23), BPD (*n* = 13), and lung ADC (*n* = 8) patients. Demographic characteristics, smoking habit and asbestos-exposure of patient groups are summarized in Table 1.

PSAP concentration was significantly higher in MPM (1773 ± 214.4 ng/mL, mean ± SEM) than in BPD (706 ± 112.4 ng/mL) samples (*p* = 0.0003) (Figure 1a). In lung ADC (1254 ± 266.7 ng/mL) samples, the biomarker level was lower and higher than in MPM and BPD samples, respectively, but the differences weren’t statistically significant (Figure 1a). SULF-1 expression level was significantly higher in MPM (0.178 ± 0.019 ng/mL) samples than in BPD (0.069 ± 0.010 ng/mL, *p* < 0.0001) and lung ADC (0.089 ± 0.017 ng/mL, *p* = 0.0062) samples (Figure 1b). Since PSAP and SULF-1 concentration values were not normally distributed, data are also presented as median, 25° percentile, and 75° percentile (Table 2).

The correlation between PSAP and SULF-1 was explored in MPM, lung ADC, and BPD patients. Both in malignant and benign PE, we found a weak-positive correlation between biomarkers (MPM, *p* = 0.022 r = 0.227; lung ADC, *p* = 0.030 r = 0.573; BPD, *p* = 0.069 r = 0.270). When all patient data were grouped a significant weak-positive correlation was observed (*p* < 0.0001 r = 0.404) (Figure 1c).

### 3.2. Diagnostic Value of PSAP and SULF-1 for MPM

The diagnostic accuracy of PSAP and SULF-1 was assessed by ROC analysis. The AUC value was calculated for each biomarker individually to determine whether everyone can discriminate between MPM and control (lung ADC and BPD) patient PE. Figure 2a,b shows ROCs of PSAP and SULF-1 differentiating between MPM and BPD pleural fluids. The AUC for PSAP was 0.853 (95% CI: 0.724–0.981, *p* = 0.0005) with 76.90% specificity and 87% sensitivity (cut-off value = 813.2 ng/mL) while the AUC for SULF-1 was 0.898 (95% CI: 0.797–0.999, *p* < 0.0001) with 92.30% specificity and 73.90% sensitivity (cut-off value = 0.12 ng/mL). Then, we investigated whether biomarker discriminative power could be increased by combining the parameters (Figure 2c). Indeed, the AUC increased to 0.916 (95% CI: 0.823–1.000, *p* < 0.0001) while specificity and sensitivity were 84.60% and 91.30%, respectively. The diagnostic power of SULF-1 to discriminate between MPM and lung ADC pleural fluids was also tested by ROC analysis resulting in an AUC value of 0.821 (95% CI: 0.658–0.984, *p* = 0.0077) with 100% specificity and 56.50% sensitivity.

### 3.3. PSAP and SULF-1 Transcript Expression in MPM and Lung ADC

In order to validate whether a differential expression of PSAP and SULF-1 between MPM and lung ADC exists, we analyzed their transcript levels in these tumors using the UALCAM data base. ULCAM (http://ualcan.path.uab.edu/cgi-bin/ualcan-res.pl; accessed on 13 October 2022) is a platform, which contains clinical and genetic data of numerous cancers from the TCGA dataset. This allows to analyze gene expression in correlation with clinical features of tumors [19,20].

The expression of PSAP transcript was found significantly downregulated in lung ADC compared to control and the normalized transcript level was lower in this tumor than in MPM. On the contrary, the expression of SULF-1 transcript was significantly upregulated in lung ADC compared to control. However, the normalized expression level was lower in lung ADC than in MPM.

Then, we examined the correlation between PSAP and SULF-1 transcript expression and MPM clinicopathological characteristic and overall survival. Neither PSAP nor SULF-1 expression showed a correlation with the tumor stage since the transcript levels were similar in all four stages. Instead, PSAP transcript level was found significantly higher in MPM epithelioid subtype than in biphasic subtype (*p* = 0.006) thus revealing a correlation between PSAP expression and histological subtype. No correlation was detected between PSAP expression level and overall survival.

The expression of SULF-1 didn’t correlate to the histological subtype. Interestingly, a correlation between SULF-1 expression and TP53 mutation was revealed with significantly higher SULF-1 transcript levels in MPM with wild-type TP53 than in the tumors with mutated TP53 (*p* = 0.002). Low/medium expression of SULF-1 was found to associate with reduced survival even though the difference between tumors with high and low/medium transcript levels was not statistically significant (*p* = 0.07).

## 4. Discussion

The positive MPM diagnosis is frequently challenging due to preclusion of obtaining a bioptic sample by thoracoscopy in those patients with frailty and compromised medical conditions. Whereas multiple cytological biomarkers can aid to differentiate MPM cells from reactive mesothelial and carcinoma cells few soluble protein biomarkers have been suggested as useful diagnostic tool in PE [9,21]. The most corroborated MPM biomarker in serum and pleural fluid is SMRP although its clinical utility as diagnostic test in isolation is hampered by the low sensitivity. The need of finding new biomarkers of diagnostic value for MPM in PE prompted us to investigate the expression levels of PSAP, which has been already validated in patient sera, and SULF-1, which has been suggested as overexpressed in MPM by transcriptomic analysis [14,17].

PSAP, which exerts numerous functional activities, is either a lysosomal 65 kDa or secreted 70 kDa glycoprotein [22]. The partially glycosylated 65 kDa isoform is the precursor of four sphingolipid activator proteins, saposins A, B, C and D, which contribute to glycosphingolipids degradation in lysosomes [23]. Therefore, deficiency of PSAP or any saposin is linked to abnormal accumulation of glycosphingolipid in lysosomes, which causes lysosomal storage diseases [24]. The fully glycosylated 70 kDa isoform, which is secreted into the extracellular space as covalently aggregated oligomers, exerts myelinotrophic and neurotrophic effects via the activation of two closely related G protein coupled receptors, GPR37 and GPR37L [25,26,27]. Various studies have also investigated the functional roles of secreted PSAP in cancer growth and metastasis reporting its effects on different cancer cell types [28,29,30,31,32,33]. Whereas this glycoprotein promotes proliferation of breast cancer cells it conversely inhibits their metastatic potential [10,28,34]. Kang et al. [30] have shown that decreased PSAP expression in human prostate cancer is associated with tumor metastasis. On the contrary, another study has suggested the involvement of PSAP in prostate cancer invasion [33]. In glioma specimens, stem cells and cell lines, PSAP has been found highly expressed even in the secreted form and the expression level seems to correlate with a poor clinical prognosis [31]. Moreover, PSAP stimulates glioma cell proliferation in vitro and regulates tumorigenesis through the toll-like receptor 4 (TLR4) associated to the nuclear factor kappa-light-chain-enhancer of activated B cells (NF-κB) signaling pathway [31]. Very recently, a glycoproteomic study has identified PSAP among other glycoproteins secreted by pancreatic ductal adenocarcinoma (PDAC) cell lines [35]. Although high PSAP expression in PDAC tissue has been found to associate with a dismal clinical prognosis the glycoprotein doesn’t seem to affect both proliferation and migration capacity of PDAC cells in vitro. However, the secreted PSAP appears to influence PDAC microenvironment reducing the number of infiltrating lymphocytes in vivo and thus promote tumor progression [35]. All these studies demonstrate that different cancer cell types express and secrete PSAP, which can affect solid tumor growth and progression by a variety of ways. In our previous study, we have shown that PSAP is secreted by a MPM cell line and high levels of this protein can be detected in serum of MPM patients [14]. In PE of MPM patients, PSAP was unsurprisingly found elevated as documented hereinafter.

SULF-1 and the closely related sulfatase 2 (SULF-2) are extracellular/cell surface sulfatases, which selectively remove 6-O-sulfate groups from glucosamine residues within the heparan sulfate (HS) chains [18]. HS is incorporated into proteoglycans (HSPGs) producing an essential component of normal tissue architecture at cell surface and within the extracellular matrix (ECM). Since these extracellular enzymes modify HSPG structure they have a role in regulating those processes, which suppress or support solid tumor growth and spread. Although these two sulfates have similar catalytic activity, they exhibit different biological functions in cancer. Whereas SULF-2 plays tumor-promoting roles in several cancer types by supporting the signaling of HS-binding growth factors, SULF-1 is generally considered an onco-suppressor albeit an oncogenic role in hepatocarcinogenesis has been reported [18,36]. Indeed, the role of SULF-1 in cancer appears to be more complex than initially thought and may correlate with the tumor type and stage. In carcinomas, SULF-1 and SULF-2 are usually downregulated and overexpressed, respectively, but their expression may change depending on tumor stage and the hypoxia level in the tumor microenvironment (TME) [18]. In ovarian cancer cell lines and primary tumors, the SULF-1 gene has been found markedly downregulated by epigenetic mechanisms [37]. Furthermore, the complete loss or reduced expression of SULF-1 has been described in cell lines derived from other cancers, which has led to suggest that SULF-1 downregulation is quite common in epithelial cancers [38]. Analysis of ovarian and breast cancer patients has shown that SULF-1 expression associates with a better prognosis validating in vitro data, which indicate a tumor suppressive role of this sulfatase [39,40]. SULF-1 and SULF-2 proteins have been reported as overexpressed in invasive PDAC surgical specimens while in four PDAC cell lines the unprocessed SULF-1 and SULF-2 protein forms were detected in one and all four detergent cell lysates, respectively [41]. Moreover, the processed SULF-2 protein was detectable in cell line culture media, but the SULF-1 wasn’t [41]. This observation may suggest that PDAC cell lines lose the ability to express and/or release SULF-1 protein due to their origin from high grade cancers and in vitro propagation. Other studies have also found higher SULF-1 mRNA levels in PDAC specimens than in normal pancreas [18]. It has been proposed that SULF-1 as well as SULF-2 are positive regulators of pancreatic cancer development through the Wnt signaling pathway [18,41]. Earlier reports have pointed out the tumor suppressive role of SULF-1 in hepatocellular carcinoma (HCC) [42,43]. The oncosuppressive effect of SULF-1 has been also supported by the more recent observation that microRNA-21 mediated silencing of SULF-1 and PTEN promotes epithelial-mesenchymal transition (EMT), proliferation, and movement of HCC cells [44]. In contrast, gene expression analysis of human HCC specimens has shown that a higher SULF-1 expression is correlated with a poor prognosis, implying an oncogenic role of this enzyme in most HCCs in vivo [45]. In support of SULF-1 oncocarcinogenetic effect in HCC, Dhanasekaran et al. have demonstrated that the enzyme activates the transforming growth factor-β (TGF-β) signaling pathway thus promoting the expression of TGF-β target genes and EMT and enhancing cell migration and invasiveness [36]. Moreover, microarray gene expression analysis of human HCC samples has confirmed that SULF-1 overexpression is associated with a poorer prognosis [36]. The expression of SULF-1 mRNA and protein has been also investigated in other solid tumors sometimes leading to discordant results that suggests the complexity to associate expression with enzyme activity [18]. In MPM specimens and cell lines, SULF-1 mRNA has been found upregulated and proposed as one of the strongest candidate cancer gene [17]. However, the predicted overexpression of SULF-1 protein wasn’t verified by Melaiu et al. [17]. Now, we show for the first time that SULF-1 protein is released in PE and its concentration is significantly high in MPM patients. Thus, our results agree with the reported upregulation of SULF-1 mRNA in this tumor although it isn’t known the role played by the enzyme in MPM development and progression.

Concerning the possible diagnostic value of PSAP and SULF-1 as biomarker in PE of MPM patients our analysis highlighted their features. First, PSAP concentration in PE is significantly higher in MPM patients than in patients with BPD suggesting its potential diagnostic role. An AUC of 0.853 suggests a good diagnostic accuracy. Predictably, the median concentration of PSAP in MPM pleural fluid is approximately 4-fold higher than that found in patient sera as consequence of the tight proximity with the tumor [14]. As for SULF-1, validated for the first time in this study, the concentration in PE was also found significantly higher in MPM patients than in BPD patients and an AUC value of 0.898 was obtained. This finding suggests a good discriminatory power of this biomarker [46]. Since SULF-1 showed higher specificity and lower sensitivity than PSAP, we reasoned that their combination using a logistic regression method could improve the predicted diagnostic performance as compared to each single biomarker. Certainly, the resulting model predicted an improvement of the discriminatory power (AUC = 0.916) together with increased specificity and sensitivity confirming that a biomarker panel performs better than the single one as diagnostic tool in PE as well as in serum [14,15,47]. To our knowledge this is second report proposing the use of PE biomarker panels as adjunct to clinical diagnosis of MPM [48].

Whereas both biomarkers may be considered good candidates as tools to discriminate between MPM and BPD their possible usefulness in distinguishing lung ADC pleural effusion is quite questionable. Firstly, no significant difference between MPM and lung ADC PSAP levels was detectable by statistical analysis. Then, SULF-1, which was found significantly higher in MPM patients than in lung ADC patients, showed some discriminative power by ROC analysis (AUC = 0.821) albeit with a low sensitivity. Indeed, both SULF-1 and PSAP have been found overexpressed in other aggressive solid tumors [18,28,31,35]. A glycoproteomic analysis of lung ADC samples and tumor-matched normal tissues has reported the overexpression of PSAP among other glycoproteins in lung ADC [49]. On the contrary, UALCAM analysis of TCGA data base showed that PSAP expression is significantly downregulated in lung ADC compared to normal tissue. Anyway, the small number of lung ADC patients enrolled in our study does not allow to precisely predict whether PSAP and SULF-1 detection in PE can discriminate between MPM and lung ADC.

The UALCAM analysis of TCGA data base showed that SULF-1 expression level doesn’t correlate with MPM stage and subtype. However, a correlation between SULF-1 expression and mutated TP53 was found. This finding is quite interesting since a lower SULF-1 expression associates with TP53 mutations and reduced survival. Recently, Quetel et al. have reported that TP53 mutations in MPM associate with poorer overall survival [50]. As for PSAP, its expression correlates with histological subtypes being the transcript levels significantly higher in the epithelioid subtype than in the biphasic subtype. The downside of the UALCAM analysis is that the number of MPM samples in the TCGA data base is only adequate for performing a comparison between epithelioid and biphasic subtypes.

All together our results show for the first time that the SULF-1 protein is likely overexpressed in MPM tissues and suggest the possible use of SULF-1 and PSAP as combined biomarkers to rapidly discriminate PEs associated to MPM from those due to BPDs when cytological findings are inconclusive. Of course, the combination of SULF-1 and PSAP with other diagnostic biomarkers can improve their possible clinical utility.

The present study has surely some weaknesses. First, a small number of PE samples for each patient groups was examined due to the exclusion of other patients for the lack of even minimal clinical and diagnostic information. Second, MPM patients were not selected based on positive or negative cytology so we cannot predict possible bias for overevaluation of biomarker diagnostic accuracy. Third, all MPMs were epithelioid subtype de facto reducing the usefulness of these biomarkers as adjunct diagnostic tools for other histological subtypes.

In conclusion, the detection of PSAP and SULF-1 in PE may be added to the adjunct tests including measurement of other soluble biomarkers besides immunocytochemistry and scans, which together support a diagnosis of MPM when invasive procedures aren’t feasible. However, further studies on larger cohorts of MPM patients and controls are required to establish a definitive diagnostic role of these putative biomarkers proposed by us.

## Figures and Tables

**Figure 1 biomedicines-10-02803-f001:**
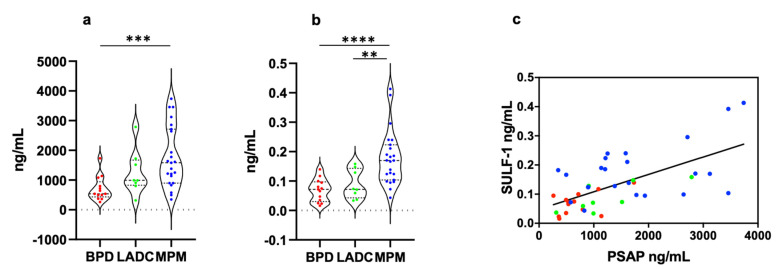
PSAP and SULF-1 concentrations in PE of MPM, lung ADC, and BPD patients. (**a**) PSAP levels were significantly higher in MPM patients than in BPD patients (*p* = 0.0003). (**b**) SULF-1 levels were significantly higher in MPM patients than in BPD (*p* < 0.0001) and lung ADC (*p* = 0.0062) patients. (**c**) Taking into account all patients, PSAP levels significantly correlated to SULF-1 expression levels (*p* < 0.0001 r = 0.404. The correlation was determined by simple logistic regression using the Likelihood-Ratio Test (LRT) (GraphPad Prism 8.0). Each data point represents mean of three independent experiments performed in duplicate. Statistical significance of the difference between groups was determined by the Mann-Whitney-Wilcoxon test. **, *p* < 0.01; ***, *p* < 0.001; ****, *p* < 0.0001. MPM, malignant pleural mesothelioma; LADC, lung adenocarcinoma; BPD, benign pleural disease.

**Figure 2 biomedicines-10-02803-f002:**
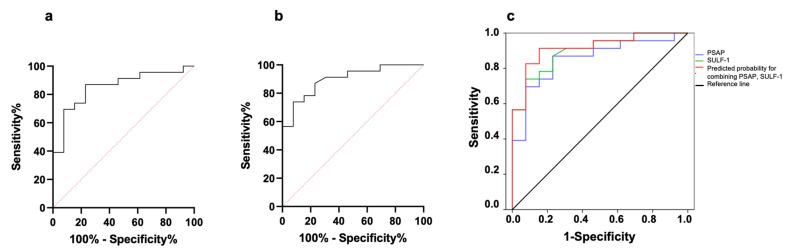
Diagnostic efficiency of individual and combined biomarkers for discriminating MPM (*n* = 23) from BPD (*n* = 13) patients. (**a**) ROC analysis showing specificity and sensitivity for PSAP. (**b**) ROC analysis showing specificity and sensitivity for SULF-1. (**c**) Individual and combined ROCs. AUC values were 0.853 and 0.898 for PSAP and SULF-1, respectively, while the combination predicted an AUC of 0.916.

**Table 1 biomedicines-10-02803-t001:** Patient demography, smoking habit, and asbestos-exposure.

		BPD	LADC	MPM
Number		13	8	23
Gender	Male Female	11 2	6 2	20 3
Age	Median Range	66 39–87	72 58–85	72 56–83
Smoke		2	3	5
Asbestos Exposure		n.a.	n.a.	5
Mesothelioma histology	Epithelial			23
Cancer histology	Adenocarcinoma		8	
Benign histology	Pleural inflammation Hyperplasia	8 5		

LADC, lung adenocarcinoma; BPD, benign pleural disease; MPM, malignant pleural mesothelioma; n.a., not available.

**Table 2 biomedicines-10-02803-t002:** Summary of PSAP and SULF-1 concentrations in patient pleural fluids.

	PSAP (ng/mL)			SULF-1 (ng/mL)		
	BPD	LADC	MPM	BPD	LADC	MPM
Median	536.6	990.2	1583	0.0715	0.0719	0.1696
25° percentile	433	829.4	898.6	0.0301	0.0426	0.1037
75° percentile	941.7	1671	2713	0.0973	0.1431	0.2235
*p*-Value	0.0003 (MPM vs. BPD)			<0.0001 (MPM vs. BPD); 0.0062 (MPM vs. LADC)		

LADC, lung adenocarcinoma; BPD, benign pleural disease; MPM, malignant pleural mesothelioma. *p*-Value of the differences between groups was determined by the MannWhitney-Wilcoxon test.

## Data Availability

The data used to support the findings of this study are available upon request to the authors.

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
