# Peer review of "Role of Prosaposin and Extracellular Sulfatase Sulf-1 Detection in Pleural Effusions as Diagnostic Biomarkers of Malignant Mesothelioma"

_biomedicines, 2022, doi:10.3390/biomedicines10112803_

Round 1
Reviewer 1 Report
Based on previous works the authors identified prosaposin and SULF-1 as potential diagnostic biomarkers for malignant pleural mesothelioma (MPM). In the present study the authors developed predictive models to show the sensitivity and specifcity of both biomarkers for MPM. In order to exemplify the discriminative power of both biomarkers, 8 patients diagnosed with lung adenocarcinoma (ADC) and 13 patients that displayed benign pleural disease (BPD) were included into the study in addition to 23 MPM patients.
The authors tackled a very imporant topic and build the study on good, previously established foundations. However, several issues need to be addressed before this article can be published.
1) The case numbers of ADC (n=8) in comparison to MPM (n=23) and BPD (n=13) is rather low. In order to proof the discriminative power of both biomarkers in MPM/ADC, the ADC case numbers should be increased to 15 at least.
2) What statistical test did the authors use for the correlation analysis displayed in figure 1C? Please outline this in the revised version of the manuscript.
3) As was done with with MPM and BPD, please show the different subtypes of ADC in table 1, since it is a very hetergenous desease with numerous histological but also molecular variants.
Reviewer 2 Report
This manuscript written by Lorenzo Zallocco, et al, is very interesting. The study design is quite simple and straightforward. They tested two proteins of PSAP and SULF-1 levels in pleural effusion including 44 patients (21 with mesothelioma, 8 with lung cancer, and 13 with non-malignant disease). They found that the elevation of these protein levels was observed in mesothelioma compared with other diseases, indicating that they may be promising biomarkers for differential diagnosis. The results support their conclusion. This study is well done and this work could be publishable if a minor revision could be made based on the comments below.
Comments:
1. The number of patients is quite small. The conclusion fro this study was drawn from a small number of patient samples. To make it more robust, this study should continue to investigate more patients. If possible, it would be more interesting to look at the levels of the proteins in different subtypes of mesothelioma.
2. Is protein level associated with overall survival of mesothelioma patients?
3. PSAP and SULF1 protein levels are increased in PE, but we don’t know where they come from tumor cells or infiltrating cells. To clarify that these proteins are produced by what type of cells, it would be helpful to do single cell RNA Seq in the future.
4. Another comment is that, to compare PSAP and SULF1 gene expression levels using mesothelioma TCGA data base, it would be doable to see if they are associated with overall survival, and histological subtypes.
5. It is also interesting to know if tumor cells themselves can produce these proteins in vitro, to detect PSAP and SULF1 concentrations in culture medium of mesothelioma cells, and gene expression by tumor cells in vitro.
6. These comments may be helpful in future study, just for your reference. I would suggest address these points in discussion.
Reviewer 3 Report
In the present manuscript, the authors evaluated the availability of PSAP and SULF-1 in pleural effusion (PE) as the diagnostic biomarkers of malignant pleural mesothelioma (MPM). Although surgical biopsy is usually needed for the accurate diagnosis of MPM, the procedure is sometimes too invasive especially for elderly patients. Therefore, it is beneficial to establish the method to diagnose MPM by thoracentesis. While I realize the importance of this manuscript, I think that the authors need to add some data before the publication in Biomedicines.
Major comments
1. Are PSAP and SULF-1 correlated with the stage of MPM? I know that the biological roles of PSAP and SULF-1 in MPM are not fully understand. However, if the value of PSAP and SULF-1 in PE is higher in patients with advanced MPM, those biomarkers are supposed to play an important role in the disease progression. If no correlation between these biomarkers and disease progression were observed, the authors at least should include the clinical stage of LADC and MPM in Table 1 because protein concentration in PE usually becomes higher according to the disease progression.
2. As the authors mentioned in the discussion, I am wondering why all the patients included in this study are cytology positive. To diagnose MPM, cytological analysis is insufficient and histological analysis is usually needed. Therefore, the patients enrolled in this study are considered to be a rare population. Most of the patients diagnosed with MPM were supposed to be cytology negative. Therefore, the authors should include the data from those MPM patients with cytology-negative. Otherwise, diagnostic value with these biomarkers would be missing.
3. The authors should include the data of other known biomarkers such as hyaluronan, SMRP, adenosine deaminase and so on. Otherwise, they cannot mention the superiority of PSAP and SULF-1 compared with these previously reported biomarkers.
Minor comment
1. The authors had better mention the history of asbestos-exposure of the enrolled patients.
Round 2
Reviewer 1 Report
The authors have improved on the previous version by including a differential expression analysis of PSAP and SULF-1 in Lung AdC and MPM based on a TCGA-based validation cohort.
To sufficiently determine the true value of PSAP and SULF-1 as diagnostic biomarkers for MPM, additional studies in larger cohorts needed.
As a stepping stone, this study is eligible for publication in Biomedicines.
Reviewer 3 Report
The revised manuscript and the authors’ response letter have addressed the previous issues that were raised.